# Consumption of Low Dose Fucoxanthin Does Not Prevent Hepatic and Adipose Inflammation and Fibrosis in Mouse Models of Diet-Induced Obesity

**DOI:** 10.3390/nu14112280

**Published:** 2022-05-29

**Authors:** Mi-Bo Kim, Minkyung Bae, Yoojin Lee, Hyunju Kang, Siqi Hu, Tho X. Pham, Young-Ki Park, Ji-Young Lee

**Affiliations:** 1Department of Nutritional Sciences, University of Connecticut, Storrs, CT 06269, USA; mi-bo.kim@uconn.edu (M.-B.K.); yoojin.lee@uconn.edu (Y.L.); hyunju.kang@uconn.edu (H.K.); siqi.hu@uconn.edu (S.H.); tho.pham@uconn.edu (T.X.P.); young-ki.park@uconn.edu (Y.-K.P.); 2Department of Food and Nutrition, Changwon National University, Changwon 51140, Korea; mkbae@changwon.ac.kr

**Keywords:** fucoxanthin, obesity, NASH, inflammation, liver fibrosis

## Abstract

Fucoxanthin (FCX) is a xanthophyll carotenoid present in brown seaweed. The goal of this study was to examine whether FCX supplementation could attenuate obesity-associated metabolic abnormalities, fibrosis, and inflammation in two diet-induced obesity (DIO) mouse models. C57BL/6J mice were fed either a high-fat/high-sucrose/high-cholesterol (HFC) diet or a high-fat/high-sucrose (HFS) diet. The former induces more severe liver injury than the latter model. In the first study, male C57BL/6J mice were fed an HFC diet, or an HFC diet containing 0.015% or 0.03% (*w*/*w*) FCX powder for 12 weeks to develop obesity-induced nonalcoholic steatohepatitis (NASH). In the second study, mice were fed an HFS diet or an HFS diet containing 0.01% FCX powder for 8 weeks. FCX did not change body weight gain and serum lipid profiles compared to the HFC or HFS controls. No significant differences were present in liver triglyceride and total cholesterol, hepatic fat accumulation, and serum alanine aminotransferase levels between control and FCX-fed mice regardless of whether they were on an HFC or HFS diet. FCX did not mitigate mRNA abundance of genes involved in lipid synthesis, cholesterol metabolism, inflammation, and fibrosis in the liver and white adipose tissue, while hepatic fatty acid β-oxidation genes were significantly elevated by FCX in both HFC and HFS feeding studies. Additionally, in the soleus muscle, FCX supplementation significantly elevated genes that regulate mitochondrial biogenesis and fatty acid β-oxidation, concomitantly increasing mitochondrial DNA copy number, compared with HFC. In summary, FCX supplementation had minor effects on hepatic and white adipose inflammation and fibrosis in two different DIO mouse models.

## 1. Introduction

Excess lipid accumulation in the liver is a crucial feature of non-alcoholic fatty liver disease (NAFLD) [1]. NAFLD encompasses a broad histological spectrum of liver damage from simple steatosis to non-alcoholic steatohepatitis (NASH), which can lead to cirrhosis [2]. Various factors, such as insulin resistance, inflammation, as well as oxidative stress, can stimulate the progression of simple steatosis into NASH, characterized by inflammation and fibrosis [3,4,5]. Therefore, in obese individuals, inhibition of inflammation and fibrosis is important for NASH prevention. 

Fucoxanthin (FCX) is a xanthophyll carotenoid abundantly present in brown seaweeds. Studies have demonstrated FCX has anti-obesity [6,7,8,9] and anti-diabetic properties [10,11]. In a mouse model of obesity, FCX reduced body weight gain by inducing genes crucial for fatty acid β-oxidation in the white adipose tissue (WAT) [6,7]. The anti-obesity effect of FCX in humans and mice is linked to increased uncoupling protein 1 (*Ucp1*) expression and browning/beiging of WAT [8,9]. In addition, FCX ameliorated insulin resistance by activating insulin receptor substrate 1 (IRS-1)/phosphatidylinositol 3-kinase (PI3K)/Akt and AMP-activated protein kinase (AMPK) signaling pathways in the liver and skeletal muscle of diabetic mice [10,11]. Furthermore, it has been shown that FCX attenuates hepatic lipid accumulation by suppressing lipogenic genes but increasing enzymes for lipolysis and β-oxidation in diet-induced obesity (DIO) mice [12,13]. Moreover, we demonstrated that FCX represses transforming growth factor β1 (TGFβ1)-induced pro-fibrogenic gene expression via downregulation of the SMA- and MAD-related protein 3 (SMAD3) pathways in a human hepatic stellate cell (HSC) line, LX-2 cells, and by suppressing the activation of quiescent primary mouse HSCs [14]. Collectively, these studies suggest that FCX may prevent the development of liver fibrosis. However, the mechanisms underlying the health benefits of FCX in preventing obesity and obesity-associated fibrosis and inflammation *in vivo* remain unclear. Therefore, we explored whether FCX can prevent the obesity-associated metabolic abnormalities, fibrosis, and inflammation in the liver, WAT, skeletal muscle, and spleen using two different DIO mouse models, i.e., those fed a diet rich in fat and sucrose with or without high cholesterol. Previously, we demonstrated that C57BL/6J mice fed a high-fat/high-sucrose (HFS) diet for 30 weeks showed NASH features but with minimal liver fibrosis [4]. We also reported that the addition of high cholesterol to the diet, i.e., high-fat/high-sucrose/high-cholesterol (HFC), tended to trigger severe liver damage with evident liver fibrosis after less than 16 weeks on the diet [15]. Using two different diets, we aimed to evaluate the potential preventive effect of FCX on mild and severe liver injury. 

## 2. Materials and Methods

### 2.1. Animal Care and Diet 

Seven-week-old male C57BL/6J mice were obtained from the Jackson Laboratory (Bar Harbor, ME, USA). Following a week of acclimatization, mice were maintained under a 12 h light/dark cycle and had free access to food and water for 12 or 8 weeks of the experimental period. In the first feeding study, mice were fed an HFC control diet (34% fat, 34% sucrose, 2.0% cholesterol, *w*/*w*, *n* = 12), an HFC diet containing 0.015% (*w*/*w*) FCX (FCX 0.015%; *n* = 12), or an HFC diet containing 0.03% FCX (FCX 0.03%; *n* = 12) for 12 weeks. In the second feeding study, male C57BL/6J mice at the age of 15 weeks were fed an HFS diet (34% fat, 35% sucrose, *w*/*w*, *n* = 12) and an HFS diet containing 0.01% FCX (FCX 0.01%; *n* = 8) for 8 weeks. FCX powder containing 10% FCX or control powder was purchased from the Flaouse Co., Ltd. (Tokyo, Japan). Therefore, to achieve 0.03% FCX in the experimental diets, 3 g of 10% FCX powder was added to formulate a 1 kg diet. The diet composition of the HFC and HFS regimes has previously been reported elsewhere [4,15]. At the end of 12- or 8-week feeding, mice were anesthetized with a mixture of ketamine (110 mg/kg) and xylazine (10 mg/kg) (Henry Schein Animal Health, Dublin, OH, USA). Blood was collected from the mice by cardiac puncture for serum preparation. Liver, epididymal WAT (eWAT), and soleus muscle were snap-frozen in liquid nitrogen and stored at −80 °C until ready to use to study gene expression or stored in 10% formalin for histological analysis. The Institutional Animal Care and Use Committee (IACUC) at the University of Connecticut reviewed and approved all the animal procedures (approved #A19-033).

### 2.2. Serum Chemistry and Liver Lipid Measurement

Serum triglyceride (TG) and total cholesterol (TC) were measured enzymatically as previously described [16]. Serum alanine transaminase (ALT) activity and glucose levels were determined using a liquid ALT (SGPT) reagent set (Pointe Scientific, Canton, MI, USA) and a liquid glucose (oxidase) reagent set (Pointe Scientific), respectively. Liver TG and TC concentrations were measured after lipid extraction from the liver using Folch’s method [17]. 

### 2.3. Histological Evaluations of Liver 

All hematoxylin and eosin (H&E) staining procedures were performed at the Connecticut Veterinary Medical Diagnostic Laboratory (CVMDL, Storrs, CT, USA). Tissue images for histological evaluations were viewed at 10× magnifications as described previously [15]. 

### 2.4. Quantitative Real-Time Polymerase Chain Reaction (qRT-PCR) 

Total RNA was extracted from the liver, soleus muscle, and eWAT. cDNA synthesis and qRT-PCR analysis using CFX96 Real-Time system (Bio-Rad, Hercules, CA, USA) were conducted and analyzed using the 2^−ΔΔCt^ method as previously published [18,19]. 

### 2.5. Splenocytes Isolation

Splenocytes isolated from the mice on experimental diets for 12 weeks were stimulated with 500 ng/mL of lipopolysaccharide (LPS) for 20 h for gene analysis [20]. 

### 2.6. Statistical Analysis

One-way analysis of variance (ANOVA), Newman Keuls post-hoc test or unpaired t-test were performed using GraphPad Prism 9.0 (GraphPad Software, La Jolla, CA, USA) to detect significant differences between groups and to check the normal distribution of the data. Data were considered significant when the *p* value was <0.05. Data are presented as mean ± SEM.

## 3. Results

### 3.1. FCX Did Not Reduce Body Weight and Serum Lipid and Glucose Levels in Obese Mice Fed an HFC Diet

Mice fed HFC control, FCX 0.015%, or FCX 0.03% diets gradually increased body weight over a 12-week feeding period (Figure 1a). After 12 weeks on experimental diets, mice fed FCX diets showed significantly higher body weight than mice on the HFC control diet (Figure 1b). eWAT weight was significantly increased in the FCX 0.03% group compared with the HFC control group (Figure 1c). Serum TC levels were significantly increased in both the FCX 0.015% and FCX 0.03% groups (Figure 1d), but serum TG and glucose levels were significantly higher in the FCX 0.015% group than the HFC control group (Figure 1e).

### 3.2. FCX Did Not Reduce Hepatic Steatosis in Obese Mice Fed an HFC Diet

We next examined the preventive effect of FCX supplementation on hepatic steatosis. Liver tissue weight showed a significant increase in the FCX 0.015% group compared to the HFC control, but there was no significant difference between the HFC control and FCX 0.03% groups (Figure 2a). FCX did not alter hepatic TG and TC content compared with the HFC control (Figure 2b). H&E staining of liver sections demonstrated a similar degree of lipid accumulation in all groups (Figure 2c). Serum ALT activity was not significantly altered by FCX consumption (Figure 2d). However, the hepatic expression levels of lipogenic genes, such as sterol regulatory element-binding protein factor 1c (*Srebf1*) and fatty acid synthase (*Fas*), were significantly elevated in both FCX groups compared to the HFC control (Figure 2e). Consistently, FCX significantly elevated the hepatic expression of genes for cholesterol metabolism, such as low-density lipoprotein receptor (*Ldlr*) and hydroxy-3-methylglutaryl-coenzyme A reductase (*Hmgcr*), compared with the HFC control, but *Srebf2* mRNA showed no significant difference between groups (Figure 2f). While hepatic expression of acyl-coenzyme A oxidase 1 (*Acox1*) involved in peroxisomal β-oxidation was significantly elevated only in the FCX 003% group compared to the HFC control, the expression levels of the mitochondrial β-oxidation gene, i.e., carnitine palmitoyltransferase 1α (*Cpt1α*), showed no significant difference between groups (Figure 2g). However, the hepatic expression of a mitochondrial biogenesis gene, peroxisome proliferator-activated receptor-γ coactivator 1α (*Ppargc1a*), was significantly increased in both FCX groups compared to the HFC control (Figure 2h).

### 3.3. FCX Did Not Prevent Fibrosis and Inflammation in the Liver of Obese Mice Fed an HFC Diet

Fibrosis and inflammation are cardinal features of NASH [4]. Hepatic mRNA levels of pro-fibrogenic genes, such as α-smooth muscle actin (*Acta2*) and collagen type 1 alpha 1 (*Col1a1*), were not significantly altered by FCX (Figure 3a). Although mRNA levels of macrophage marker adhesion G protein-coupled receptor E1 (*Adgre1*), also known as F4/80, and tumor necrosis factor α (*Tnf*) were not significantly altered by FCX, interleukin (*Il)1b* mRNA was significantly increased in both FCX 0.015% and FCX 0.03% groups compared with the HFC control (Figure 3b,c). However, hepatic mRNA expression of *Il6* was significantly increased in only the FCX 0.015% group compared to the HFC group control. 

Splenic monocytes may play a critical role in hepatic inflammation [20]. Therefore, the LPS reactivity of the splenocytes isolated from mice fed experimental diets was assessed *ex vivo*. FCX did not alter spleen weight and the splenic expression of *Il1b*, *Il6*, and *Tnf*, regardless of LPS stimulation (Appendix A). 

### 3.4. FCX Had Minimal Effects on the Expression of Genes Involved in Inflammation, Fibrosis, and β-Oxidation in the eWAT of Obese Mice Fed an HFC Diet

As body and eWAT weights were elevated by FCX supplementation, we further investigated the effect of FCX on fibrosis and inflammation therein. While the FCX 0.03% group showed a significant increase in the expression levels of *Adgre1* and *Tnf* compared to the HFC control, *Il1b* and CC-chemokine ligand 2 (*Ccl2*) mRNA abundance was not significantly altered by FCX supplementation (Figure 4a,b). The eWAT expression levels of pro-fibrogenic genes, such as *Col1a1* and *Col6a3*, were not significantly altered by FCX, but *Col6a1* mRNA was significantly elevated in FCX fed groups compared to the HFC control (Figure 4c). *Acox1* mRNA was significantly elevated in the eWAT of FCX 0.03% group compared to the HFC control, but *Cpt1α* and *Ppargc1a* expressions were not significantly changed by FCX supplementation (Figure 4d). 

### 3.5. FCX Elevated Expression of Genes Related to Energy Utilization in Soleus Muscle of Obese Mice Fed an HFC Diet

FCX is known to reduce the risk of metabolic syndrome by increasing energy expenditure [21]. Thus, we evaluated the expression of genes responsible for mitochondrial biogenesis and lipid oxidation in soleus muscle. The expression levels of *Cpt1α* and *Acox1* were significantly increased in the FCX 0.03% group compared to the HFC group (Figure 5a). While *Ucp2* mRNA was significantly higher in the FCX 0.03% group than in the other groups, *Ucp3* mRNA was significantly diminished by FCX 0.015%. The expression of mitochondrial biogenesis genes, e.g., mitochondria transcription factor A (*Tfam*), estrogen-related receptor α (*Esrrα*), and *Ppargc1a*, was significantly elevated in FCX-fed groups compared to the HFC control (Figure 5b). Furthermore, mitochondrial DNA was significantly higher in the soleus muscle of the FCX-fed groups compared to the HFC group (Figure 5c). 

### 3.6. FCX Supplementation Increased Body Weight and Serum TC Levels in Mice Fed an HFS Diet 

Studies have demonstrated that FCX consumption in the range of 0.02–0.2% (*w*/*w*) has beneficial effects on liver health and obesity [12,22,23]. However, we did not observe significant effects of FCX at 0.015 or 0.03% (*w*/*w*) on obesity, fibrosis, and inflammation in an HFC diet-induced obesity mouse model. Brown seaweed is known to contain 0.01 to 0.07% of FCX by dry weight [24]. Therefore, we further examined whether FCX supplementation at a lower level (FCX 0.01%, *w*/*w*) would attenuate obesity-associated inflammation and fibrosis in HFS diet-induced obesity mice. The FCX 0.01% group showed significantly higher body weight than the HFS control from 1 week on the experimental diets (Figure 6a). After eight weeks on the experimental diets, the body weights of the FCX 0.01% group were significantly higher than the HFS group, but eWAT and spleen weights were not significantly different between the groups (Figure 6b–d). While the FCX 0.01% group had significantly higher serum TC concentration than the HFC control, there was no significant change in serum TG and glucose concentrations by FCX consumption (Figure 6e,f).

### 3.7. FCX Did Not Alter Hepatic Steatosis and Fibrosis, but Increased the Expression of Genes-Related to Energy Utilization in the Soleus Muscle of Mice Fed an HFS Diet

FCX 0.01% group did not significantly alter the liver weight and serum levels of ALT compared to HFS control (Figure 7a,b). Liver TG, but not TC, contents were significantly elevated in the FCX 0.01% group compared to the HFS control (Figure 7c). Mice fed FCX 0.01% had a significantly higher hepatic expression of *Srebf1* and *Fas* than the HFS control (Table 1). In addition, the hepatic expression of *Adgre1* and *Col1a1* was significantly higher in the FCX 0.01% group than in the HFS control, while there was no significant difference in *Acta2* expression between the FCX 0.01% group and the HFS control. However, the hepatic expression of *Acox1* and *Ppargc1a* was significantly elevated in the FCX 0.01% group compared to the HFS control, which was consistent with the results in the HFC-fed mice. Although *Cpt1α* and *Ppargc1a* did not differ between the two groups, *Ucp2* and *Acox1* were significantly higher in the FCX 0.01% group in soleus muscle. Genes related to mitochondrial biogenesis and quantity, i.e., *Ppargc1a*, *Esrra*, and *Tfam*, were not significantly different between the groups. 

## 4. Discussion

NAFLD is an increasingly important global health issue. However, pharmacotherapy for treating NAFLD has not been approved, and there are few treatment options available for this disease [25]. Effective lifestyle therapies, including a healthy diet and physical activity, are good strategies for the prevention or treatment of NAFLD [26]. In the current study, we used two DIO mice, one of which develops advanced NASH, e.g., the HFC diet model, to examine the preventive effect of FCX on fibrosis and inflammation in the liver and eWAT. We recently demonstrated that the progression of metabolic, inflammatory, and fibrotic features of HFC and HFS diet mouse models occurred by performing an *in vivo* study [4,15]. This result showed that the mouse models developed as well-characterized mouse models of NASH with a strong manifestation of liver fibrosis. Despite well established control mice, we found that FCX supplementation had a minimal impact on body weight, serum lipid profiles, and hepatic fat accumulation, as well as fibrosis and inflammation in the liver and eWAT, although FCX led to the induction of genes crucial for mitochondrial biogenesis and fatty acid β-oxidation and increased mitochondrial copy number in the soleus muscle of the two DIO mouse models. 

Studies have shown that FCX supplementation reduced body weight gain by enhancing fatty acid β-oxidation in the liver and adipose tissue and increased serum total and apolipoprotein B-containing lipoprotein cholesterol levels by lowering the activity of two key hepatic cholesterol-regulating enzymes, i.e., 3-hydroxy-3-methylglutaryl coenzyme A reductase and acyl coenzyme A: cholesterol acyltransferase, in obese mice [9,12]. However, we did not observe a similar effect of FCX in our two DIO mouse models. This discrepancy may be related to different diet compositions used in the studies. In our study, mice were fed an HFC diet containing 34% fat, 34% sucrose, and 2.0% cholesterol for twelve weeks. However, in other studies mice were fed a diet containing 20% fat and 50% sucrose for six weeks [9,12]. As FCX can be easily degraded by gastric acid in the gastrointestinal (GI) tract [27] and it is a hydrophobic substance, other components in experimental diets can alter the absorption and biotransformation of FCX, impacting its biological effect. FCX shows a higher anti-obesity effect when it is administered together with medium-chain triacylglycerols (MCT) than FCX alone in diabetic/obese KK-Ay mice with increased expression of *Ucp1* in WAT [28]. Moreover, in the same mice, the mixture of FCX (0.1%) and fish oil (6.9%) showed a reduction in blood glucose levels and WAT weight to a similar extent as 0.2% FCX alone, suggesting that the combination of FCX and fish oil led was almost twice as effective as FCX alone [6]. Furthermore, a combination of FCX and conjugated linoleic acid reduced body weight gain and liver weight more than FCX alone in high-fat diet-induced obese rats [6,29]. It should be noted, however, that the health benefits reported in these studies may be compounded by oils present in experimental diets because it is well known that MCT, fish oil, and conjugated linoleic acid can reduce body weight by increasing energy expenditure and decreasing serum triglycerides and lipogenesis by stimulating lipid oxidation in animals and humans [30,31,32]. In the present study, we formulated control and FCX powders for diets so that the only difference between the two formulations was FCX to avoid any confounding effects of other diet ingredients. In addition, differences in FCX dosages used in other studies from the present study likely contributed to the discrepancy in findings. When mice were fed a high-fat diet supplemented with FCX (0.05% or 0.2%, *w*/*w*) for six weeks, FCX decreased hepatic triglyceride and cholesterol accumulation by inhibition of hepatic lipogenesis concomitantly with elevated hepatic lipolysis. [22]. Moreover, diabetic/obese KK-Ay mice fed 0.1% FCX (*w*/*w*) for three weeks exhibited significant reductions in hepatic lipid content. However, consumption of an AIN-93G diet containing FCX (0.025% or 0.05%, *w*/*w*) for five weeks did not alter the liver weight and hepatic lipid profiles in C57BL/6J mice [33]. Higher FCX doses than our study have been used in other studies, some of which are not achievable in humans. Therefore, careful evaluation of the health benefits of FCX using physiological FCX doses without confounding factors in experimental diets is needed to validate if FCX consumption can be beneficial or not. 

In the present study, FCX supplementation increased serum TC levels in both mouse models, consistent with the findings in other studies. When FCX at doses of 1,000 and 2,000 mg/kg was orally administrated to ICR mice for 30 days, plasma TC concentration was significantly increased without changes in plasma levels of TG and ALT and histological abnormalities of the liver [34]. In addition, consumption of 0.2% (*w*/*w*) FCX in KK-Ay obese mice for four weeks increased serum TC and high-density lipoprotein (HDL)-cholesterol concentrations [35]. Serum cholesterol homeostasis is primarily maintained by hepatic low density lipoprotein receptor (LDLR), which takes up LDL from the circulation, lowering serum LDL cholesterol concentrations [36]. In the current study, although hepatic expression of *Ldlr* was significantly induced by FCX, serum TC was higher than in control groups, which may be related to elevated HDL cholesterol. The role of FCX on HDL metabolism needs to be investigated using a proper animal model, such as human apolipoprotein A-I transgenic mice that exhibit heterogenous HDL subpopulations, similar to those of humans [37]. 

We recently demonstrated that FCX prevented TGFβ1-stimulated pro-fibrogenic gene expression by inhibiting SMA- and MAD-related protein activation in LX-2 cells and primary human HSCs [14]. FCX attenuated the activation of quiescent HSCs, supporting an inhibitory effect of FCX on fibrogenesis. Furthermore, the anti-inflammatory effects of FCX have also been reported in several obese mouse models [38,39]. Chronic tissue damage induces inflammation, which is an essential factor for fibrosis development in the eWAT and liver [40]. In high-fat diet-induced obese mice, FCX supplementation for four weeks reduced plasma levels of IL-1β and TNFα [38]. When diabetic/obese KK-Ay mice were fed an AIN-93G diet containing FCX (0.1%, *w*/*w*) for three weeks, FCX significantly down-regulated pro-inflammatory mediators, such as *Tnf* and monocyte chemoattractant protein 1, in eWAT [39]. However, in the present study, FCX supplementation did not attenuate diet-induced fibrosis and inflammation in the liver and eWAT. In fact, there was increased expression of *Adgre1* and *Tnf* in the WAT of HFC supplemented with 0.03% FCX. It is unclear what caused this discrepancy, but it may be related to different mouse strains, diet composition, duration of feeding study, FCX supplementation level, and other dietary ingredients present in experimental diets. As discussed above, our FCX supplementation levels were less than that used in other studies. Our FCX supplementation levels are achievable in humans, but the FCX used in some other studies are supraphysiological levels. Another possibility is that our diet triggers severe fibrosis and inflammation in the liver and eWAT, which may not be counteracted by FCX at the levels used in our studies. 

It was notable that soleus muscle from mice fed FCX showed a significant increase in the expression of genes related to mitochondrial biogenesis and fatty acid β-oxidation, along with elevated mitochondrial copy number, only in the HFC study but not in the HFS study. It is not clear what caused this discrepancy between the two diets. However, it is feasible that cholesterol abundant in the HFC diet might cause more metabolic stress, eliciting a mitochondrial stress response in skeletal muscle. This FCX effect on mitochondria in the HFC study is consistent with the findings from other studies using different mouse models. In *db/db* diabetic mice, FCX supplementation (0.2% or 0.4%, *w*/*w*) for six weeks significantly increased the hepatic protein level of PPARα, phosphorylated acetyl-coenzyme A carboxylase, and CPT-1, which might contribute to the reduction in plasma TG and TC concentrations [10]. FCX supplementation (0.2%, *w*/*w*) in a high-sucrose (50% sucrose) or high-fat (30% fat) diet for five weeks significantly elevated the expression of genes involved in mitochondrial metabolism and fatty acid β-oxidation in inguinal WAT and eWAT, resulting in increased metabolic rate and reduced adipose mass in mice [41]. Moreover, 0.2% FCX supplementation for two weeks in diabetic/obese KK-Ay mice increased *Ppargc1a* expression in both soleus and extensor digitorum longus muscle, which was linked to decreased blood glucose and insulin levels [11]. Enhanced mitochondrial function can decrease fat accumulation and reactive oxygen species generation and attenuate chronic inflammation to prevent NASH development [42,43]. Thus, it is worth evaluating the effect of FCX on mitochondrial functions in muscle in the future. 

## 5. Conclusions

In conclusion, we demonstrated that FCX at supplementation levels of 0.01, 0.15, and 0.03% (*w*/*w*) did not prevent diet-induced fibrosis and inflammation in the liver and eWAT of two DIO mouse models. However, FCX enhanced genes involved in mitochondrial biogenesis and fatty acid β-oxidation, which could elevate the capacity of skeletal muscle. Therefore, further studies are warranted to determine the role of FCX supplementation on energy expenditure and metabolic rate by enhancing mitochondrial function in the context of metabolic diseases.

## Figures and Tables

**Figure 1 nutrients-14-02280-f001:**
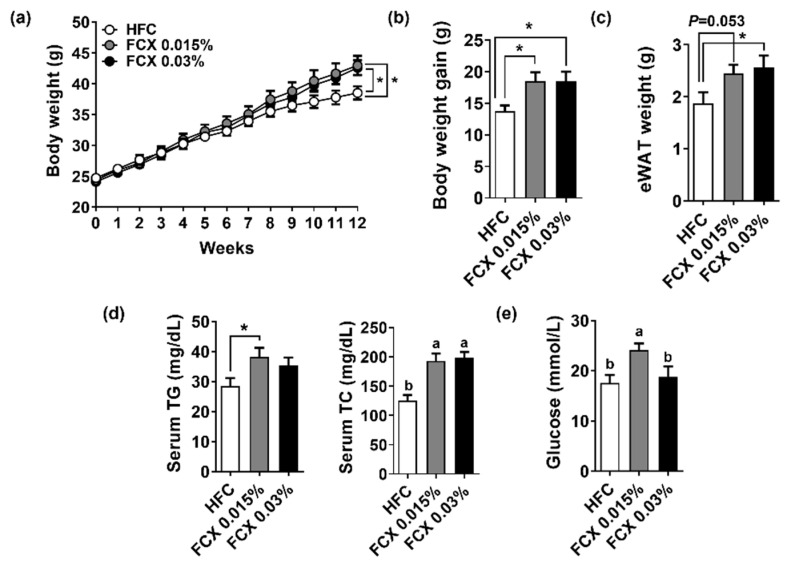
Effect of FCX on body and tissue weight changes and serum levels of lipids and glucose in DIO mice fed HFC control, FCX 0.015%, or FCX 0.03% diet for 12 weeks. (**a**) Body weight change throughout the 12 weeks of the experiment diet. (**b**) Body weight gain. (**c**) eWAT weight. (**d**) Serum TG and TC concentrations. (**e**) Serum fasting glucose levels. *n* = 8–11 per group. Mean ± SEM. Bars without sharing the same letter are significantly different (*p <* 0.05). * *p* < 0.05 between two groups.

**Figure 2 nutrients-14-02280-f002:**
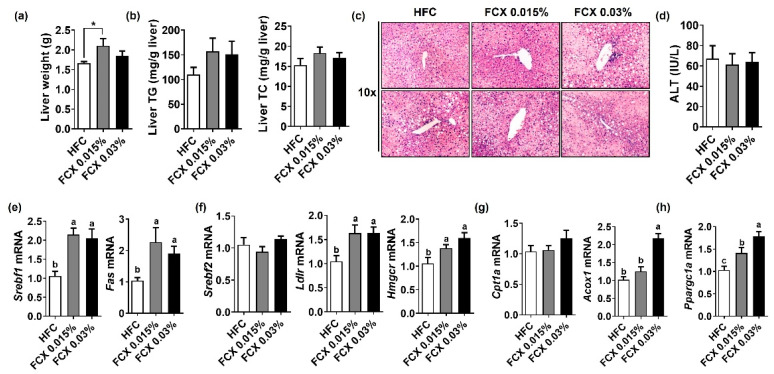
Effect of FCX on liver steatosis in DIO mice fed HFC control, FCX 0.015%, or FCX 0.03% for 12 weeks. (**a**) Liver weight. (**b**) Liver TG and TC levels. *n* = 9–11 per group. (**c**) Liver sections stained with H&E. *n* = 8 per group. (**d**) Serum ALT levels. *n* = 8–11 per group. Hepatic mRNA levels of (**e**) lipogenic genes, (**f**) cholesterol metabolism genes, (**g**) β-oxidation genes, and (**h**) mitochondrial biogenesis-related gene. *n* = 9–11 per group. Mean ± SEM. Bars without sharing the same letter are significantly different (*p <* 0.05). * *p* < 0.05 between two groups.

**Figure 3 nutrients-14-02280-f003:**
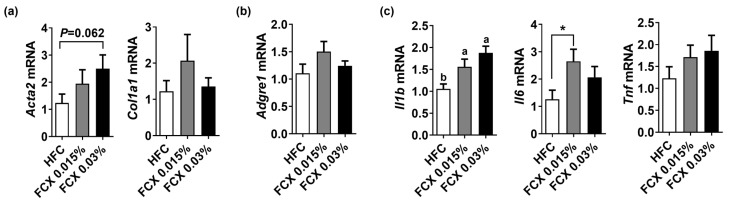
Effect of FCX on the expression of pro-fibrogenic and inflammation genes in the liver of DIO mice fed HFC control, FCX 0.015%, or FCX 0.03% for 12 weeks. Hepatic mRNA levels of (**a**) pro-fibrogenic genes, (**b**) macrophage marker, and (**c**) inflammation genes. *n* = 9–11 per group. Mean ± SEM. Bars without sharing the same letter are significantly different (*p <* 0.05). * *p* < 0.05 between two groups.

**Figure 4 nutrients-14-02280-f004:**
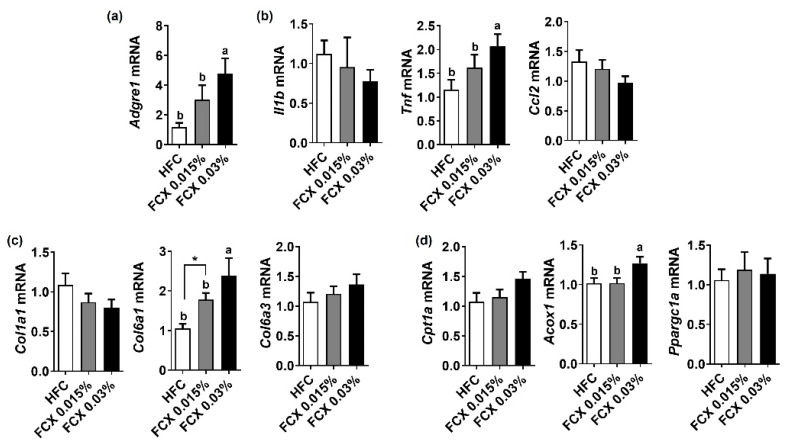
Effect of FCX on the expression of inflammation, profibrogenic, and β-oxidation genes in the eWAT of DIO mice fed HFC control, FCX 0.015%, or FCX 0.03% for 12 weeks. eWAT mRNA expression levels of (**a**) macrophage marker F4/80, (**b**) inflammation genes, (**c**) pro-fibrogenic genes, and (**d**) β-oxidation genes. *n* = 8–11 per group. Mean ± SEM. Bars without sharing the same letter are significantly different (*p <* 0.05). * *p* < 0.05 between two groups.

**Figure 5 nutrients-14-02280-f005:**
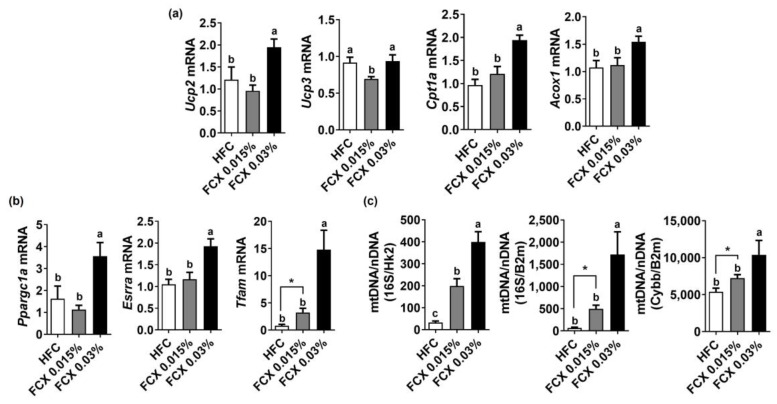
Effect of FCX on the expression of energy-utilization-related genes in the soleus muscle of DIO mice fed HFC, FCX 0.015%, or FCX 0.03% for 12 weeks. The expression of genes related to (**a**) β-oxidation and (**b**) mitochondrial biogenesis in soleus muscle. (**c**) Mitochondria DNA copy number in soleus muscle. *n* = 6–11 per group. Mean ± SEM. Bars without sharing the same letter are significantly different (*p <* 0.05). * *p* < 0.05 between two groups.

**Figure 6 nutrients-14-02280-f006:**
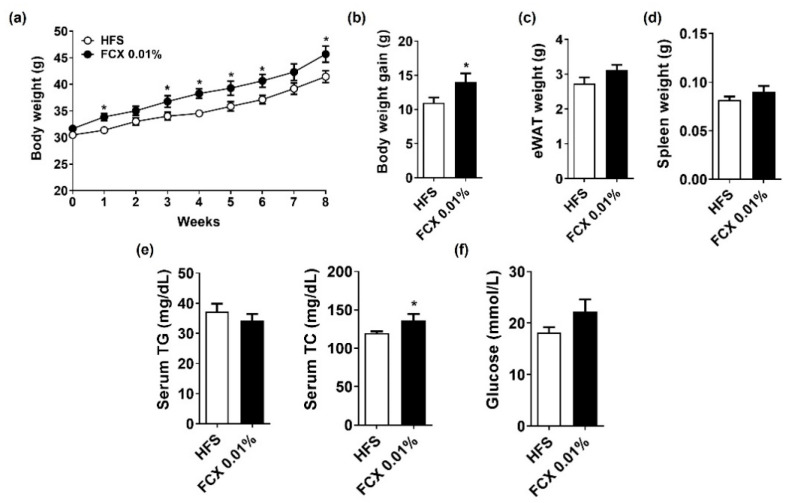
Effect of FCX on body and tissue weight changes and serum levels of lipid profiles and glucose in DIO mice fed HFS control or FCX 0.01% diet for 8 weeks. (**a**) Body weight of mice through 8 weeks of the experimental period. (**b**) Body weight gain. (**c**) eWAT weight. (**d**) Spleen weight. (**e**) Serum TG and TC concentrations. (**f**) Serum fasting glucose levels. *n* = 7–11 per group. Mean ± SEM. *, *p* < 0.05 between two groups.

**Figure 7 nutrients-14-02280-f007:**
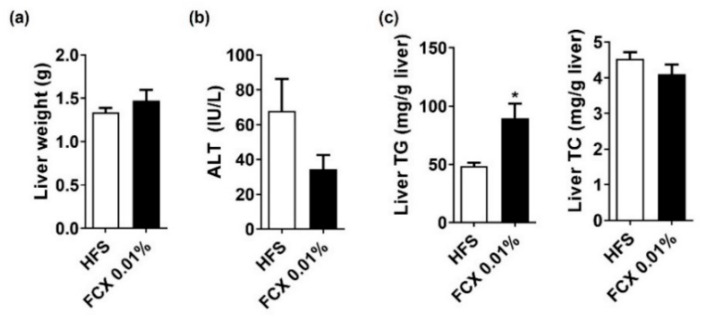
Effect of FCX on hepatic steatosis and fibrosis in DIO mice fed HFS control or FCX 0.01% for 8 weeks. (**a**) Liver weight. (**b**) Serum ALT levels. (**c**) Liver TG and TC levels. *n* = 9–11 per group. Data mean ± SEM. * *p* < 0.05 between two groups.

**Table 1 nutrients-14-02280-t001:** Gene analysis in the liver and soleus muscle of C57BL/6J mice fed HFS or FCX 0.01% for 8 weeks.

Tissue	Gene	HFS	FCX 0.01%
Liver	*Srebf1*	1.10 ± 0.16	1.95 ± 0.15 *
	*Fas*	1.01 ± 0.06	1.51 ± 0.17 *
	*Adgre1*	1.03 ± 0.08	1.53 ± 0.19 *
	*Acta2*	1.11 ± 0.17	1.42 ± 0.36
	*Col1a1*	1.09 ± 0.13	1.88 ± 0.24 *
	*Cpt1a*	1.01 ± 0.04	1.16 ± 0.06
	*Acox1*	1.01 ± 0.04	1.22 ± 0.08 *
	*Ppargc1a*	1.03 ± 0.08	1.50 ± 0.11 *
Soleus muscle	*Ucp2*	1.01 ± 0.04	1.32 ± 0.15 *
	*Ucp3*	1.04 ± 0.10	1.41 ± 0.15
	*Cpt1a*	1.10 ± 0.14	1.49 ± 0.27
	*Acox1*	1.03 ± 0.07	1.72 ± 0.32 *
	*Ppargc1a*	1.13 ± 0.17	0.97 ± 0.21
	*Esrra*	1.05 ± 0.09	1.54 ± 0.31
	*Tfam*	1.02 ± 0.06	1.19 ± 0.11

Data are expressed as relative expression to HFS control and mean ± SEM (*n* = 7–11 per group). * Significantly different from HFS in a row (*p* < 0.05).

## Data Availability

Not applicable.

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
