# Peer review of "Consumption of Low Dose Fucoxanthin Does Not Prevent Hepatic and Adipose Inflammation and Fibrosis in Mouse Models of Diet-Induced Obesity"

_nutrients, 2022, doi:10.3390/nu14112280_

Round 1

Reviewer 1 Report

The current study is on a topic of general interest to the readers of the journal, where obesity related complications are severe with high global rates of mortality and morbidity. However, the description of some very important points was inadequate, so I have several questions and concerns.

Major comments

  • The title is so confusing as “Fucoxanthin Supplementation” does not indicate that Fucoxanthin is incorporated in the diet, rather it implies that it was administered as a supplement.
  • The aim of the study is not clear, the authors stated that: “the underlying mechanisms of FCX in preventing obesity and obesity-associated inflammation and fibrosis in vivo remains unclear”, despite that the authors referred to previous studies that elaborated its in vitro and in vivo hepatoprotective mechanisms

Moreover, the authors mentioned “Therefore, we sought to investigate whether FCX supplementation can prevent the obesity-associated metabolic abnormalities, inflammation, and fibrosis in two different DIO mouse models”. However, it came clear in the discussion (lines 277-287) that they aimed to study the effects of diet composition on FCX.

  • FCX when orally administered (up to 400 mg/kg) was able to induce hepatoprotection against NAFLD as elaborated in previous studies (Shih et al., 2021; Zheng et al., 2019). So why do authors expect that such a small proportion of FCX (0.015-0.013%) could induce hepatoprotection, especially that the authors mentioned that “As FCX can be easily degraded by gastric acid in the gastrointestinal (GI) tract and it is a hydrophobic substance, other components in experimental diets can alter absorption and biotransformation of FCX, impacting its biological effect”.

References

Shih, P.H., Shiue, S.J., Chen, C.N., Cheng, S.W., Lin, H.Y., Wu, L.W., Wu, M.S., 2021. Fucoidan and Fucoxanthin Attenuate Hepatic Steatosis and Inflammation of NAFLD through Modulation of Leptin/Adiponectin Axis. Mar. Drugs 19. https://doi.org/10.3390/MD19030148

Zheng, J., Tian, X., Zhang, W., Zheng, P., Huang, F., Ding, G., Yang, Z., 2019. Protective Effects of Fucoxanthin against Alcoholic Liver Injury by Activation of Nrf2-Mediated Antioxidant Defense and Inhibition of TLR4-Mediated Inflammation. Mar. Drugs 17. https://doi.org/10.3390/MD17100552

  • Also, incorporation of drugs/compounds in diet is not of clinical applicability
  • The authors should have applied Masson Trichrome dye in their histopathological examinations, as H&E is not enough for fibrosis examination and evaluation.

Author Response

Reviewer 1

The current study is on a topic of general interest to the readers of the journal, where obesity related complications are severe with high global rates of mortality and morbidity. However, the description of some very important points was inadequate, so I have several questions and concerns.

  1. The title is so confusing as “Fucoxanthin Supplementation” does not indicate that Fucoxanthin is incorporated in the diet, rather it implies that it was administered as a supplement.

We changed it to “Dietary fucoxanthin consumption” in title.

  1. The aim of the study is not clear, the authors stated that: “the underlying mechanisms of FCX in preventing obesity and obesity-associated inflammation and fibrosis in vivo remains unclear”, despite that the authors referred to previous studies that elaborated its in vitro and in vivo hepatoprotective mechanisms.

Although previous studies have demonstrated beneficial effects on liver health in vitro and in vivo, little is known about the role of FCX in hepatic and adipose fibrosis in DIO mouse models. We used a HFC diet model in this study that manifests NASH with advanced liver fibrosis, which has never been investigated before. Therefore, we believe this study is novel.

  1. Moreover, the authors mentioned “Therefore, we sought to investigate whether FCX supplementation can prevent the obesity-associated metabolic abnormalities, inflammation, and fibrosis in two different DIO mouse models”. However, it came clear in the discussion (lines 277-287) that they aimed to study the effects of diet composition on FCX.

For clarification, we did not study the effects of diet composition on biological effects of FCX. We brought up a potential role of diet composition to explain discrepancy observed in our and other studies as to FCX effects.

  1. FCX when orally administered (up to 400 mg/kg) was able to induce hepatoprotection against NAFLD as elaborated in previous studies (Shih et al., 2021; Zheng et al., 2019). So why do authors expect that such a small proportion of FCX (0.015-0.013%) could induce hepatoprotection, especially that the authors mentioned that “As FCX can be easily degraded by gastric acid in the gastrointestinal (GI) tract and it is a hydrophobic substance, other components in experimental diets can alter absorption and biotransformation of FCX, impacting its biological effect”.

Our investigation was exploring the effect of FCX when it is consumed as diet. In some studies, supraphysiological doses of FCX were used. We used FCX supplementation at a lower level (FCX 0.015-0.013%, w/w), as this is achievable in humans through diet.

  1. Also, incorporation of drugs/compounds in diet is not of clinical applicability.

We respectfully disagree with this comment. There are various dietary supplements of a single compound in the market, supporting the clinical applicability of these compounds.

  1. The authors should have applied Masson Trichrome dye in their histopathological examinations, as H&E is not enough for fibrosis examination and evaluation.

We agree with the reviewer’s comments that the Masson Trichrome staining will be helpful for the evaluation of fibrosis. However, we did not observe any significant changes in the expression of fibrogenic genes. Based on our previous experience, the expression of hepatic fibrogenic genes changes in parallel to liver fibrosis. Therefore, we do not think the histological analysis of liver fibrosis will generate any meaningful findings.

Reviewer 2 Report

This is an interesting study by Kim et al. regarding the role of fucoxanthin (FCX) supplementation on hepatic lipid content, inflammation, and fibrosis, in a mouse model of NAFLD/NASH. The study uses much more reasonable concentrations of FCX compared to previous studies that have shown a protective effect of this compound on mouse liver steatosis and fibrosis. However, that study is characterized by several shortcomings.  

Major issues 
1. The study lacks protein verification of the multiple qPCR results reported. Only minor, non-specific HE images are provided. 
2. Please report if all data had normal distribution and if not use the proper statistical test. 
3. The method for the measurement of tissue triglycerides and total cholesterol is not described 
4. A method for the quantitation of tissue lipid content on HE staining is not described. The authors should use a lipid-specific staining (e.g. oil-red-o) and a proper quantitation method. 
5. The authors should also provide fibrosis specific staining images (e.g. mason's trichrome) with proper quantitation or use a protein-based method (e.g. tissue hydroxyproline or col1a1 western-blots) to clearly support their point of no FCX effect on diet-induced liver fibrosis 
6. Results reported in the relevant text and in Figure 2g are not in agreement. Please clarify that only the high concentration of FCX has an effect on Acox1. 
7. Similarly, results reported in the relevant text and in Figure 3c are not in agreement. Please clarify that only the low concentration of FCX has an effect on IL6. 
8. Lines 168-172. No supplemental Figures were included in the provided material to evaluate. 
9. The researchers did not include a control group fed with a normal diet to verify the effectiveness of their model (that it indeed induced liver steatosis and fibrosis). This is a major shortcoming. 
10. Results presented in Table 1 should be further explained. Specifically, the authors should address the HFS column and how these results were obtained relatively to HFS control.
11. The utility of studying gene expression in the soleus muscle should be further clarified early in the text. 

Minor issues 
1. The authors need to comment on the supraphysiological concentration of fat on their diet (34%) and its potential effect on their results. They briefly address this issue in the discussion (lines 345-347)

Author Response

This is an interesting study by Kim et al. regarding the role of fucoxanthin (FCX) supplementation on hepatic lipid content, inflammation, and fibrosis, in a mouse model of NAFLD/NASH. The study uses much more reasonable concentrations of FCX compared to previous studies that have shown a protective effect of this compound on mouse liver steatosis and fibrosis. However, that study is characterized by several shortcomings. 

Major issues

  1. The study lacks protein verification of the multiple qPCR results reported. Only minor, non-specific HE images are provided.

We measured the expression of genes related to lipid metabolism, inflammation and fibrosis in the liver and WAT of two mouse models. To supplement the gene data, we also conducted histological and biochemical measurements, along with phenotyping parameters. Based on our previous experiences, we did not think measuring proteins would change our conclusions.

  1. Please report if all data had normal distribution and if not use the proper statistical test.

       We checked the normality of data when performing statistical analyses.

  1. The method for the measurement of tissue triglycerides and total cholesterol is not described.

Liver TG and TC concentrations were measured after lipid extraction from the liver using modified Folch's method ( Carr, T. P., et al. (1993). "Enzymatic determination of triglyceride, free cholesterol, and total cholesterol in tissue lipid extracts." Clinical biochemistry 26(1): 39-42.), which was reflected in the materials and methods section.  

  1. A method for the quantitation of tissue lipid content on HE staining is not described. The authors should use a lipid-specific staining (e.g. oil-red-o) and a proper quantitation method.

We did not conduct the quantification of lipid content on HE staining, as they are qualitative dataset. To complement this data, we also measured liver TG and TC contents.

  1. The authors should also provide fibrosis specific staining images (e.g. mason's trichrome) with proper quantitation or use a protein-based method (e.g. tissue hydroxyproline or col1a1 western-blots) to clearly support their point of no FCX effect on diet-induced liver fibrosis.

We agree with the reviewer’s comments that the Masson Trichrome staining will be helpful for the evaluation of fibrosis. However, we did not observe any significant changes in the expression of fibrogenic genes. Based on our previous experience, the expression of hepatic fibrogenic genes changes in parallel to liver fibrosis. Therefore, we do not think the histological analysis of liver fibrosis will generate any meaningful findings.

  1. Results reported in the relevant text and in Figure 2g are not in agreement. Please clarify that only the high concentration of FCX has an effect on Acox1.

Thank you for pointing this out. We revised it.

  1. Similarly, results reported in the relevant text and in Figure 3c are not in agreement. Please clarify that only the low concentration of FCX has an effect on IL6.

Thank you for pointing this out. We revised it.

  1. Lines 168-172. No supplemental Figures were included in the provided material to evaluate.

The supplemental figure was submitted.

  1. The researchers did not include a control group fed with a normal diet to verify the effectiveness of their model (that it indeed induced liver steatosis and fibrosis). This is a major shortcoming.

We have extensive experience with two mouse models used in this study. For instance, we recently demonstrated that the progression of metabolic, inflammatory and fibrotic features of HFC diet mouse model was monitored by performing in vivo time-course study ( Kim, M.-B., et al. (2020). "Comprehensive characterization of metabolic, inflammatory and fibrotic changes in a mouse model of diet-derived nonalcoholic steatohepatitis." The Journal of nutritional biochemistry 85: 108463). This result showed that mouse models were developed as a well-characterized mouse model of nonalcoholic steatohepatitis (NASH) with a strong manifestation of liver fibrosis.

  1. Results presented in Table 1 should be further explained. Specifically, the authors should address the HFS column and how these results were obtained relatively to HFS control.

       We provided the information in qRT-PCR of the materials and methods section.

  1. The utility of studying gene expression in the soleus muscle should be further clarified early in the text.

       As per the reviewer’s suggestion, we clarified in line 215-216.  

Minor issues

  1. The authors need to comment on the supraphysiological concentration of fat on their diet (34%) and its potential effect on their results. They briefly address this issue in the discussion (lines 345-347).

We added this aspect.  

Round 2

Reviewer 1 Report

Authors has responded to the requested comments

Author Response

Reviewer 1 was satisfied with our responses. 

Reviewer 2 Report

The authors have adequately responded to some of my previous review points. Only points 3,4,6,7, and 8 were addressed correctly. The rest of my major points have not been appropriately handled, and the authors have avoided providing critical information. 

previous point 1. No protein verification is provided and the explanation is not adequate. 

previous point 2. The normal distribution of data is not reported in the text.

previous point 5. Since there was a significant increase in the expression of Col1a1 in the FCX 0.01% group this is not a proper answer. 

previous point 9. The missing control group in an experiment that can present significant biological variability is critical information. 

previous point 10. The provided information is not satisfactory. Why does HFS normalized expression range in all genes from 1.01 to 1.13?  

Author Response

Reviewer 2

  1. previous point 1. No protein verification is provided and the explanation is not adequate. 

As we responded previously, we performed histological and biochemical analyses to supplement gene data. These phenotypic evaluations are more physiologically relevant than proteins, as changes in protein levels do not necessarily reflect phenotypic changes. We hope the review shares the same view with us. 

  1. previous point 2. The normal distribution of data is not reported in the text.

We added to the materials and methods section.

  1. previous point 5. Since there was a significant increase in the expression of Col1a1 in the FCX 0.01% group this is not a proper answer. 

Although Col1a1 mRNA was increased by FCX 0.01%, there was no significant change in another marker of liver fibrosis, i.e., Acta2. Therefore, based on our experience, we do not think the histological analysis of liver fibrosis will generate any meaningful findings.

  1. previous point 9. The missing control group in an experiment that can present significant biological variability is critical information. 

We respectfully disagree with the reviewer that no healthy diet control is not a shortcoming of this study. The primary goal of this study is to evaluate the effect of FCX on obesity-related dysfunctions. Based on our previously published studies, we know the HF diets used in this study trigger the dysfunctions in obesity. Therefore, the primary comparison should be HF vs. HF + FCX to achieve the goal of this study.     

  1. previous point 10. The provided information is not satisfactory. Why does HFS normalized expression range in all genes from 1.01 to 1.13?  

The information was added to Table 1, as requested.
